# A Comparative Genomic and Phylogenetic Investigation of the Xenobiotic Metabolism Enzymes of Cytochrome P450 in Elephants Shows Loss in CYP2E and CYP4A

**DOI:** 10.3390/ani13121939

**Published:** 2023-06-09

**Authors:** Kanami Watanabe, Mitsuki Kondo, Yoshinori Ikenaka, Shouta M. M. Nakayama, Mayumi Ishizuka

**Affiliations:** 1Laboratory of Toxicology, Department of Environmental Veterinary Science, Faculty of Veterinary Medicine, Hokkaido University, Sapporo 060-0818, Japanshoutanakayama0219@gmail.com (S.M.M.N.); 2National Institute for Environmental Studies (NIES) Biodiversity Division, Ecological Risk Assessment and Control Section, Tsukuba 305-8506, Japan; 3Water Research Group, Unit for Environmental Sciences and Management, North-West University, Potchefstroom 2520, South Africa; 4Translational Research Unit, Veterinary Teaching Hospital, Faculty of Veterinary Medicine, Hokkaido University, Sapporo 060-0818, Japan; 5One Health Research Center, Hokkaido University, Sapporo 060-0818, Japan; 6Biomedical Sciences Department, School of Veterinary Medicine, The University of Zambia, P.O. Box 32379, Lusaka 10101, Zambia

**Keywords:** cytochrome P450, genome database, phylogenetics tree, Asian elephant, African elephant, Afrotheria, cytochrome P450 2E, cytochrome P450 4A

## Abstract

**Simple Summary:**

Our study investigates a specific genetic loss and duplication in elephants by comparing genomic data from 12 different mammalian species. We created a synteny map and phylogenetic tree of the selected species, which suggested that elephant CYP1–4 genes have undergone independent duplication, particularly in the CYP2A, CYP2C, and CYP3A genes, a unique cluster found only in elephants. In contrast, CYP2E and CYP4A might have decayed to a state of genetic dysfunction (pseudogene). These results suggest the need for further investigations into elephant genomics to evaluate the species’ metabolic mechanisms.

**Abstract:**

Cytochrome P450 is an important enzyme that metabolizes a variety of chemicals, including exogenous substances, such as drugs and environmental chemicals, and endogenous substances, such as steroids, fatty acids, and cholesterol. Some CYPs show interspecific differences in terms of genetic variation. As little is known about the mechanisms of elephant metabolism, we carried out a comparative genomic and phylogenetic analysis of CYP in elephants. Our results suggest that elephant CYP genes have undergone independent duplication, particularly in the CYP2A, CYP2C, and CYP3A genes, a unique cluster specific to elephant species. However, while CYP2E and CYP4A were conserved in other Afrotheria taxa, their decay in elephants resulted in genetic dysfunction (pseudogene). These findings outline several remarkable characteristics of elephant CYP1–4 genes and provide new insights into elephant xenobiotic metabolism. Further functional investigations are necessary to characterize elephant CYP, including expression patterns and interactions with drugs and sensitivities to other chemicals.

## 1. Introduction

Elephants play a vital role in maintaining the ecological balance and biodiversity of their habitats. As “ecosystem engineers”, their behaviors shape the landscape and create habitats for other species. The loss of elephants due to poisoning can have consequences for other animals, such as scavenger species (e.g., vultures and hyenas), and the ecosystem [1,2,3]. There have been several cases of suggested elephant poisoning, including the mass death of nearly 450 elephants in the Republic of Botswana between 2020 and 2021 [4,5] and an increase in the reports of Asian elephants dying from ingested pesticides and agricultural chemicals throughout India [6]. To effectively address the issue of elephant poisoning, it is crucial to understand the mechanisms of xenobiotic metabolism in elephants. Additionally, dosages used for medical treatment of captive elephants have often been extrapolated from data on horses based on anatomical similarities [7,8]; however, differences in xenobiotic metabolisms pose a risk of over- or underdosing [9,10,11]. Thus, detailed research investigating the metabolic mechanisms of xenobiotics, such as drugs and environmental chemicals, is necessary in elephants.

Cytochrome P450 (CYP) enzymes are responsible for the metabolism of a wide range of endogenous and exogenous compounds, including drugs, toxins, and environmental pollutants [12,13,14,15,16,17]. The substrate selectivity of CYP enzymes, which are mainly expressed in the liver, refers to their ability to metabolize specific substrates or classes of compounds. CYP enzymes can be broadly classified into families and subfamilies based on their sequence similarities. Each family or subfamily can have unique substrate selectivity enabling the metabolism of a variety of chemicals [12]. Of the 57 human CYPs, isoforms belonging to the CYP1-4 family are involved in the metabolism of approximately 80% of human clinical drugs [13,15,18], as well as other xenobiotics, including environmental chemicals and endogenous substances such as steroids, fatty acids, and cholesterol [12,13,16]. Additionally, some CYPs show interspecific differences in catalytic activity, suggesting that a comparative genomic and phylogenetic analysis of CYPs is necessary to characterize and distinguish enzyme forms between taxa. Furthermore, isoform numbers in CYPs are important for evaluating their function. In general, if the number of isoforms expands in a particular CYP, this indicates that the gene has been duplicated to adapt to an environment or diet change that has likely exposed the gene to a particular substrate. However, a low or deficient number of isoforms indicates that the gene is no longer exposed to a particular substrate, and its function may have selectively degenerated [19,20,21].

In this study, we aim to classify the existing xenobiotic metabolic types of CYP, specifically CYP1A, 2A-E, 3A, and 4A, in African elephants and Asian elephants. We achieved this by comparing annotated gene data for each gene assembly of three closely related species from Afrotheria (the aardvark, manatee, and Cape elephant shrew) and various well-known mammals (the horse, rat, dog, cat, pig, cattle, and human) whose xenobiotics have been studied and whose sequence annotations are known. We examined within- and between-species relationships by comparing the isoform numbers, phylogenetic tree, and synteny maps of the highly divergent or deficient cytochrome P450 gene family.

## 2. Materials and Methods

### 2.1. Data Retrieval and Isoform Numbers in CYP1–4 Sequences

To determine the number of mammalian CYP1–4 family genes, we retrieved the CYP1–4 sequences of 12 different species: the African elephant (*Loxodonta africana*), Asian elephant (*Elephas maximus*), aardvark (*Orycteropus afer*), West Indian manatee (*Trichechus manatus*), Cape elephant shrew (*Elephantulus edwardii*), domestic horse (*Equus ferus caballus*), Norway rat (*Rattus norvegicus*), domestic dog (*Canis lupus familiaris*), domestic cat (*Felis catus*), domestic pig (*Sus scrofa domesticus*), domestic cattle (*Bos taurus*), and human (*Homo sapiens*) by comparing annotated gene data for each assembly (the assembly and annotation name are provided in Appendix A). All genomic sequence information was retrieved from the National Center for Biotechnology Information (NCBI) using Basic Local Alignment Search Tool (BLAST) searches by querying the human sequences for CYP1A1, 2A13, 2B6, 2C19, 2D6, 2E1, 3A4, and 4A11. Other CYP2 and CYP3 subfamily genes such as CYP2J, 2R, 2S, 2T, 2U, and 2Ws are generally known as biosynthesis-type or unknown substrate isoforms [22,23,24,25,26], and were not included in the analyses. BLAST searches were conducted in the nucleotide database (nr/nt) for each species using BLASTN (optimized for similar sequences) with default parameters.

For further confirmation of the isoform numbers, the conserved CYP coding loci in each species were checked using NCBI’s genome data viewer (https://www.ncbi.nlm.nih.gov/genome/gdv/, (accessed on 1 May 2022)). For example, a conserved region of “AXL Receptor Tyrosine Kinase (AXL)” and the “Egl-9 family hypoxia inducible factor 2 (EGNL2)” [27] were identified to confirm the existence of existing annotated CYP2As genes. Sequences that were annotated as pseudogenes, coding sequences shorter than 50% of the average size, and sequences that contained stop codons in the first 50% of the region were all considered partial genes and excluded from these analyses. We counted the remaining existing sequences to evaluate the isoform numbers in CYP1–4 of the examined species. A phylogenetic tree at organism level (TimeTree 5 [28]) was generated together with the isoform numbers result.

### 2.2. Synteny Analysis of CYP Genes

NCBI’s MapViewer (http://www.ncbi.nlm.nih.gov/projects/mapview/, (accessed on 1 May 2022)) and SnapGene software (http://www.snapgene.com, (accessed on 1 May 2022)) were used to visualize chromosomal synteny maps for comparing Afrotheria species. Synteny maps provide a framework for illustrating the conservation of homologous genes and gene order identified between genomes of different species. A conserved region was identified following this method for a CYP region.

### 2.3. Phylogenetic Analysis of Mammalian CYP Genes

The gene sequences used are listed in Appendix A. The deduced amino acid sequences were aligned using MEGA X (Molecular Evolutionary Genetics Analysis) (https://www.megasoftware.net/, (accessed on 1 May 2022)) and MUSCLE (Multiple Sequence Comparison by Log-Expectation). Human CYP1A1 and CYP1A2 were used as an outgroup for the CYP2A, 2C, and 3A phylogenetic tree analysis. All positions with gaps or missing data were eliminated. The aligned sequence was used for model selection based on Bayesian Information Criterion (BIC) score, and a JTT + G model was used for construction of maximum likelihood trees (bootstrapping = 100).

### 2.4. Further Pseudogene Confirmation Analysis

Sequences that were not detected in the transcriptome database for elephants during the NCBI BLAST searches were further analyzed to confirm whether they were unannotated or pseudogenes. To this end, reference sequence of human CYP2E1 (Gene ID: 1571/NM_000773) and CYP4A11 (Gene ID: 1579/NM_000778), and manatee CYP2E1 (LOC101343596/XM_012555150 and LOC101344375/XM_012555152) and CYP4A11 (LOC101345950/XM_012554256 and LOC101346203/XM_004371774) mRNAs were used as a query sequence for the NCBI BLAST searches. BLAST searches were conducted on the whole genome sequence (WGS) database for each elephant species using BLASTN (optimized for similar sequences).

Human sequences and sequences found in Afrotheria were aligned using MUSCLE together with the sequences retrieved from the African and Asian elephant database. The CYP2E sequences from elephants annotated with exon1–9 segments were obtained separately, and segments were manually aligned to the reference sequence of CYP2E1. Each exon in the elephant’s genomes was aligned to a human coding sequence (CDS) exon (Exon1: 1–177, Exon2: 178–337, Exon3: 338–487, Exon4: 488–648, Exon5: 649–825, Exon6: 826–967, Exon7: 968–1155, Exon8: 1156–1297, and Exon9: 1298–1714). For CYP4A, several genes were annotated as pseudogenes. All pseudogenes in African elephant CYP4As (LOC100674332, LOC100674052, LOC100673772, LOC111751011, and LOC100673478) and Asian elephant CYP4As (LOC126071316, LOC126071317, LOC126071318, and LOC126071727) as well as a “low-quality protein” gene in Asian elephant (LOC126071726) were aligned together. Finally, we detected stop codons or any mutations in the aligned sequences of CYP2E and CYP4A.

## 3. Results

### 3.1. CYP Isoform Numbers

Isoform numbers of 12 selected mammal species were analyzed and compared (Figure 1). The average number of isoforms of each subfamily among examined species was also calculated for comparison. The results uncovered species-specific differences. Little variation was observed in the CYP1A, 2B, 2D, and 2E subfamilies. CYP2A, 2C, and 3A had undergone frequent duplication events, with higher numbers of duplications in African and Asian elephants. Some isoforms were not annotated or were missing: CYP2E was not detected by a BLAST search using a query for human CYP2E1 in either elephant species. CYP4A was conserved as a low-quality protein in Asian elephants and was detected as a pseudogenized gene in African elephants. These results suggest that CYP2E might be defective in the Tethytheria clade (which includes elephants and manatees), while CYP4A was only defective in African and Asian elephants. Although comparisons between species’ isoform numbers indicate a significant expansion in herbivores such as horses, in elephants, only half the number of copies was detected.

### 3.2. Phylogeny Tree and Synteny Map of CYP2A, 2C, and 3A 

Phylogenetic tree analysis showed that CYP2As from both elephant species were placed in the same elephant-specific clade (Figure 2B). The coding loci of CYP2As among mammals were highly conserved between AXL and EGNL2. We found that CYP2As from close relatives, such as the aardvark and Cape elephant shrew, were placed closer to the clade of other mammals than the elephant clade. The manatee’s CYP2A (XM_004390843/LOC101351737) was found in NCBI, but it was excluded from the analysis based on our analysis sequence, making up less than 50% of the entire length of the coding region. In the synteny maps of Afrotheria (Figure 2A), elephants show a greater length of the CYP2A coding region and significant gene expansion, including pseudogenes, compared to the other two species in the clade (aardvark and Cape elephant shrew). In particular, the region length between the AXL and EGLN2 genes in both elephants was about 2-fold longer than in the aardvark and Cape elephant shrew. Although the entire length of the gene cluster may not directly reflect the numbers of gene duplication/loss, it does indicate that these loci are variable among Afrotheria species. A comparison of African and Asian elephants revealed more complex duplications and deletions in Asian elephants. A total of 13 genes, including pseudogenes, were found in the common conserved region in African elephants, and 16 genes were found in that of Asian elephants. These results hint at the presence of different metabolic mechanisms in these two species.

A phylogenetic analysis and synteny comparison of the CYP2C cluster region was conducted next (Figure 3). The coding loci of CYP2Cs were highly conserved between “helicase lymphoid specific (HELLS)” and the “PDZ and LIM domain 1 (PDLIM1)” (Figure 3A), as was also reported in a previous study [19,29]. The phylogenetic tree showed that CYP2C of elephants can be separated into five different groups; CYP2C19 (clade 1), another group of CYP2C19 (clade 2), CYP2C42 (clade 3), CYP2C21 (clade 4), and CYP2C23 (clade 5). The clades can be further grouped into three broad categories. The first category includes clade 1, clade 2, and clade 3, which duplicated and developed almost exclusively in Afrotheria, except for CYP2C6 in rats. The second category, containing clade 4, shows that genes were conserved in cattle, horses, dogs, Cape elephant shrews, and manatees. Finally, the divergence of clade 5 happened earlier and was located far from the other CYP2Cs. This gene was also conserved and showed orthologs in cattle, horses, pigs, rats, and dogs (Figure 2B). The results indicate that 2C19 genes in elephants are characteristic genes that are only conserved in the two elephant species and manatees. The results further suggest that the divergence of CYP2C19 might have happened after the separation of the Tethytheria clade. CYP2C21 were found in other herbivores, such as horse and cattle. A BLAST search for the CYP2C23 gene was conducted using a query for rat CYP2C23 to determine the existence of the gene in all targeted animals. Except for dog, the CYP2C23 of all species in which it was detected (African and Asian elephant, horse, cattle, and pig) was conserved in the coding loci between “ER lipid raft associated 1 (ERLIN1)” and “carboxypeptidase N subunit 1(CPN1)”. Further, synteny maps comparing Afrotheria illustrate that the coding loci between HELLS and PDLIM1 show higher duplication and loss events in Asian elephants than in African elephants. We did not find completely connected cluster loci for the aardvark and Cape elephant shrew genomes. It should be noted that the complement of the conserved region between HELLS and PDLIM1 genes shown here may be incomplete as some genes may not have been localized during the synteny analysis.

Lastly, CYP3A phylogenetic and synteny map analyses were conducted (Figure 4). CYP3As in Afrotheria were conserved between the gene of “sidekick cell adhesion molecule 1 (SDK1)” and “forkhead box K1 (FOXK1)” (Figure 4A), but in other species was conserved between “Zinc finger and SCAN domain containing25 (ZSCAN25)” and “Tripartitemotif containing 4 (TRIM4)”. Elephant CYP3A was classified into two groups, with group 1 genes, including clade 1 and clade 2, clustered within clades of other mammals, and elephant-specific replication was observed within this clade. Group 2, including clade 3, formed a cluster of only Afrotheria species, which was isolated from other mammalian clades. Additionally, the synteny map illustrates that the CYP3A coding loci between SDK1 and FOXK1 were conserved among Afrotheria (Figure 4A). In contrast to previous reports, the synteny map showed common isoform variation in Asian and African elephants. The conserved region between SDK1 and FOXK1 genes shown in manatees was incomplete, and some genes could not be localized during the synteny analysis.

### 3.3. Pseudogene Confirmation of CYP2E and 4A

CYP2E and CYP4A were searched using BLAST, similar to other genes. However, these genes were either not detected or were reported as low-quality proteins in elephants. Since BLAST searches for elephant CYP2E in the nucleotide database (nr/nt) did not return any annotated sequences, a further search using the WGS database was conducted. For CYP4As, five African elephant genes (LOC100673478, LOC111751011, LOC100673772, LOC100674052, and LOC100674332) and four Asian elephant genes (LOC126071316, LOC126071317, LOC126071318, and LOC126071727) were annotated as pseudogenes, and one Asian elephant gene (LOC126071726) was annotated as a low-quality protein. Further confirmation was performed as described above. A stop codon or frame shift was found in CYP2E (Figure 5). Stop codons were found in Exon2 (#343-345nt) and Exon3 (#421-423nt). The position of the stop codons was identical for African and Asian elephants, indicating CYP2E pseudogenization may have occurred before the two species separated. For CYP4As, several pseudogenes were conserved in the Asian and African elephant stop codons detected in all genes. The remaining gene, “LOC126071726”, was conserved as a low-quality protein. However, we also confirmed the existence of a stop codon in the Exon2 (#310-312nt) region in both African and Asian elephants. Other losses of the CYP4A gene in African and Asian elephants were visualized and compared to confirm pseudogene in Appendix A. Additionally, the stop codon found in both CYP2E and CYP4A was located on the loci at less than 50% of the full length of the gene sequence. CYP2E is highly conserved as an ortholog gene between “synaptonemal complex central element protein 1 (SYCE1)” and “scavenger receptor family member expressed on T cells 1 (SCART1)” (Figure 6A). However, while SYCE1 genes were detected in both elephants and manatees, SCART1 was absent in these taxa. Similarly, a comparison of synteny maps of CYP4As indicated a fully conserved region between “EF-hand calcium binding domain 14 (EFcbd14)” and “STIL centriolar assembly protein (STIL)” in both African and Asian elephant (Figure 6B).

## 4. Discussion

In this study, we utilized the genome assembly data of multiple mammals to compare and evaluate the characteristic of elephant CYP1-4 gene function. Notably, the CYP2A, 2C, and 3A genes showed high duplication in both African and Asian elephants, while CYP2E and CYP4A were the pseudogenes.

### 4.1. High Duplicated Enzymes

In humans, the CYP2A family includes CYP2A6, 2A7, and 2A13 [13]. Among three enzymes, CYP2A6 predominantly expresses in the liver, which accounts for 3.4% of the metabolism of clinically used drugs which may include coumarin, nicotine, disulfiram, fadrozole, halothane, osigamone, methoxyflurane, pilocarpine, promazine, and valproic acid [13]. Apart from humans, an example of an animal with high duplication of CYP2A is the woodrat (*Neotoma bryanti*, *Neotoma lepida*). The high level of CYP2A gene duplication in the woodrat is thought to be due to its adaptation to an herbivorous diet, which exposes it to plant-derived toxic xenobiotics [30]. The Asian elephant exhibits a similar level of duplication to humans and rats. Elephants have been described as “generalized feeders” in the wild, and they are exposed to various xenobiotics by consuming more than 100 kg of plant material (encompassing up to 400 species of plants), soil, and water daily to maintain their massive bodies [31,32,33,34,35]. The expansion of metabolic enzymes, such as CYP2As, could be explained by these dietary behaviors. However, CYP2A analysis results clearly indicate a difference in isoform number between African (counted number of 13 genes including pseudogenes) and Asian elephants (16 genes), with the African species showing slightly fewer isoform numbers. This difference between the species may reflect adaptations to different ecological niches, including their habitats and the availability of food sources in the wild. Some studies suggest African elephants primarily occur in savannah and forest habitats, where they consume a variety of grasses, shrubs, leaves, and tree bark [36,37,38]. In contrast, Asian elephants prefer forested habitats and rely more heavily on the consumption of tree bark, roots, and other woody vegetation [39,40,41]. Asian elephants achieve higher digestion coefficients for dry matter (36–53 vs. 22–42%), hemicellulose (53 vs. 40%), and cellulose (47 vs. 37%) than African elephants when fed comparable diets [8,42,43,44]. Additionally, Asian elephants retain food longer in the digestive tract relative to their body size, and they have adapted to a variety of dietary strategies due to higher digestibility [35]. However, diets may still show substantial variation due to location or seasonal effects [31,33,35,36], making it difficult to identify the exact plants or xenobiotics that may have driven an expansion of metabolic enzymes.

CYP2Cs also showed high duplication in elephants. The human CYP2C subfamily consists of four highly homologous genes (CYP2C18, CYP2C19, CYP2C9, and CYP2C8) that are involved in the metabolism of ~16% of medically employed drugs [13]. The CYP2Cs are known to metabolize a wide variety of chemicals, including xenobiotic drugs, such as non-steroidal anti-inflammatory drugs (NSAIDs), anticoagulants, anticonvulsants, antidepressants [45], and endogenous compounds, such as arachidonic acid and some steroids [46,47,48]. The isoform numbers also differed between African (9 genes) and Asian (6 genes) elephants. Adaptive expansion of CYP2Cs has been reported in koalas (*Phascolarctos cinereus*) and allows the moderation of its unique diet of eucalyptus foliage, which is toxic to most mammals [49]. The above explanation may elucidate the comparatively high expansion in Asian elephants since their adaptation to a unique diet that is different from that of African elephants. CYP2C23 are other genes found in elephants, several herbivores, and carnivores. The CYP2C23 of rats and mice are known as arachidonic acid or eicosanoid metabolizing CYPs [50,51], considered more like endogenous-metabolizing isoforms, which may play a similar role in these taxa [19].

Enzymes in the CYP3A subfamily play a major role in the metabolism of ~30% of clinically used drugs from almost all therapeutic categories [13,15,16]. Examples of drugs metabolized by CYP3A include sedatives (midazolam and diazepam), antiarrhythmics (amiodarone, quinidine, and lidocaine), antidepressives (amitriptyline and imipramine), immunosuppressants (cyclosporin A, tacrolimus, and macrolide), and antibiotics (erythromycin) [13,52]. In addition to drugs, CYP3A is involved in the oxidation of a variety of endogenous substrates, such as steroids, bile acids, and retinoic acid [45]. In both African and Asian elephants, the copy number of this gene was similar. CYP3A4 has several large binding pockets where substrates or other molecules can bind and undergo chemical reactions. The ability of CYP3As to bind a large and structurally diverse set of compounds enables the enzymes to metabolize a broad range of xenobiotics [53,54]. Our results suggest that one enzyme clade, located close to the CYP3As of other mammals, may have a similar bulky substrate variety and activity, whereas other clades at a greater distance may have different features. Research using a recombinant assay may further clarify the substrate variety of each isoform.

### 4.2. Loss Pseudogene Enzymes 

Pseudogenization events in CYP2E and 4A were indicated by loss-of-function mutations, which introduced one or more stop codons in sequences (Figure 5, Appendix A). CYP2E is an important enzyme responsible for the metabolism of alcohol and compounds of low molecular weight, such as acetaminophen, isoniazid, and some anesthetics [13,55]. One of its main functions is the metabolism of ethanol. In relation to this function, several studies have shown that elephants have a low tolerance for alcohol and may exhibit behavioral changes after consuming even small amounts of ethanol [56]. Previous research conducted in mammals to identify genetic variation in aldehyde dehydrogenase IV (ALDH 7), the enzyme that converts ethanol into a metabolizable form, revealed multiple pseudogenization events in the ALDH 7 gene of elephants [57]. Incidences of African elephants becoming intoxicated after consuming fermenting the fallen fruit of the marula tree (*Sclerocarya birrea*) [56] may be partially explained by both ALDH7 and a CYP2E gene deficiency (this study). In most mammals, CYP2E is highly conserved as an ortholog gene and is conserved between the SYCE1 and SCART1 genes. However, in the present study, while the SYCE1 gene was detected in the elephant and manatee, the SCART1 gene was not. We did, however, detect a pseudogenized CYP2E gene in this region in the African elephant, suggesting that the region may have previously had a CYP2E gene that subsequently degenerated. Phylogenetic analysis in a previous study revealed that human CYP2Cs or CYP2E diverged from avian CYP2C23, indicating that mammalian CYP2E separated from CYP2Cs after the evolutionary separation of birds and mammals [58]. We showed that other Afrotheria species, such as the Cape elephant shrew and aardvark, possess CYP2E genes, a fact that indicates that pseudogenization events occurred selectively after these taxa separated into the Tethytheria clade. Although the loss was identified in both elephants and manatees, the stop codon and frame-shifted position were not conserved between elephants and manatees, indicating that details in when and how the gene became a pseudogene differ among species.

Dietary habits can have an impact on unstable CYP gene variation that has undergone rapid birth–death evolution (frequent gene duplications and losses), while endogenous substrate genes are generally stable and show less variation [59]. However, depending on their physiology and homeostasis mechanism, these stable genes could also have evolved into dysfunctional genes. For example, previous studies reported the loss of enzymes in the bile acid synthesis pathway in elephants and manatees. Bile acids are important for absorbing nutrients, and are mostly composed of cholic and chenodeoxycholic acid. In elephants and manatees, both cholic acid and chenodeoxycholic acid are, however, absent; instead, they contain bile acid alcohols [60,61]. Differences in bile composition between elephants and manatees were associated with the loss of CYP8B1, SLC27A5, and ACOX2 genes, which are connected to the bile acid biosynthesis pathway [61]. Thus, the selective loss of certain genes could be the result of species-specific physiology, such as endogenous substrate metabolism. Our study also showed a loss of the CYP4A gene, an enzyme involved in both endogenous and exogenous metabolism, such as steroids and fatty acids [16,62]. Unlike other genes reported to be defective in both manatees and elephants, CYP4A was found to be defective only in elephants. Further confirmation is needed to conclude gene loss, whether other alternative metabolic enzymes for the genes for which we identified as pseudogenes exist, and if any one of them may have a mechanism for xenobiotic metabolism.

Recent studies indicate that CYP2E1 may also play a role in stimulating responses to oxidative stress by increasing the production of reactive oxygen species (ROS). In breast cancer cells, increased CYP2E1 expression is associated with increased ROS levels, suggesting that CYP2E1 may play a role in promoting cancer cell progression, metastasis, and stage advance [63,64]. Elephants are known to have a very low incidence of cancer, despite their large number of cells and long lifespan. One explanation given for this is that elephants have 20 copies of the cancer suppressor gene p53 [65], whereas humans have only 1 copy. Interestingly, there is some evidence of an interaction between CYP2E1 and p53, suggesting that p53 may regulate the expression of CYP2E1 in the liver [63,64]. Some studies have shown that p53 modulates the toxicity of certain drugs, including those metabolized by CYP2E1.

Furthermore, studies investigating the physiological dysfunction of the CYP2El gene in knockout mice showed no observed phenotypic abnormalities [66]. These findings indicate that CYP2E1 is not essential for normal development or physiological homeostasis, and the loss of CYP2E1 does not affect the expression of other CYPs. In contrast, the knockout mice with disrupted Cyp4a10 and Cyp4a14 genes exhibited a hypertensive phenotype [67,68,69]. These studies revealed that the targeted genes, Cyp4a10 and Cyp4a14, were not directly responsible for the hypertensive phenotype, but instead regulated the expression of other P450 genes involved in hypertension. Although the deficiency of CYP2E1 did not show an effect on physiological characteristics, it has been associated with nonalcoholic fatty liver disease (NAFLD). NAFLD is a disease associated with obesity, diabetes, and metabolic syndrome, and the progression of NAFLD can lead to the development of nonalcoholic steatohepatitis (NASH). CYP2E1 and CYP4A, which are abundant enzymes, are associated with this NASH [70]. CYP2E1 has been reported to play a crucial role in promoting the development of NASH by initiating lipid peroxidation through increased production of reactive species. However, studies using Cyp2E1 knockout mice showed that NASH development was not prevented, and there was no reduction in microsomal NADPH-dependent lipid peroxidation. Meanwhile, upregulation of CYP4A10 and CYP4A14 was observed in these mice, indicating their potential compensatory roles. To further understand the role of CYP4A in this pathway, Cyp4A knockout mice were used, revealing the activation of CYP4As as an alternative pathway for producing reactive oxygen species (ROS) [71]. These results suggest that the absence of both CYP2E1 and CYP4A in elephants may be associated with a lower generation of ROS and a lower incidence of certain diseases such as NASH [70,71].

The study provides initial insights into the presence and genetic characteristics of CYP subfamilies in African and Asian elephants, as well as other mammalian species. However, it is important to acknowledge the limitations that gene structure may not directly indicate functional enzymatic activity or metabolic function. Future research should focus on experimental investigations to determine tissue-specific expression patterns and metabolic activities of these CYPs. The study was also constrained by limitations in the quality of genome assemblies, which made it challenging to analyze certain CYP genes. Nonetheless, the findings highlight the presence of elephant-specific metabolic mechanisms and caution against extrapolating physiological characteristics across species in drug use. Continued evaluation of this genomic data will play a crucial role in understanding species differences and further investigating the roles of these CYPs in processes such as xenobiotic metabolism, environmental adaptation, and other physiological functions in elephants.

## 5. Conclusions

Although there is room for further study, our results indicate the possible existence of chemical substances and drugs that elephants cannot metabolize and of elephant-specific detoxification and metabolism mechanisms. Deficiency of CYP2E has not been reported in other mammalian taxa; only now, in this study, is it reported in elephants. Therefore, we recommend combining our results with functional analysis of both in vivo (pharmacokinetics and pharmacodynamics) and in vitro (metabolic activity) elephant models.

## Figures and Tables

**Figure 1 animals-13-01939-f001:**
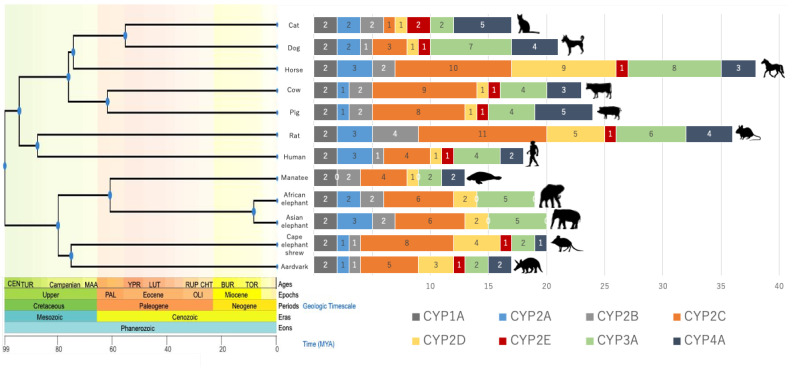
Number of isoforms of the CYP1–4 families in 12 selected mammal species. The number of genes of CYP1A, 2A, 2B, 2C, 2D, 2E, 3A, and 4A are labeled in the bar graph. Any gene annotated as a “low-quality protein” was determined using the methods described in the text. The phylogenetic tree was created with TimeTree 5 [28].

**Figure 2 animals-13-01939-f002:**
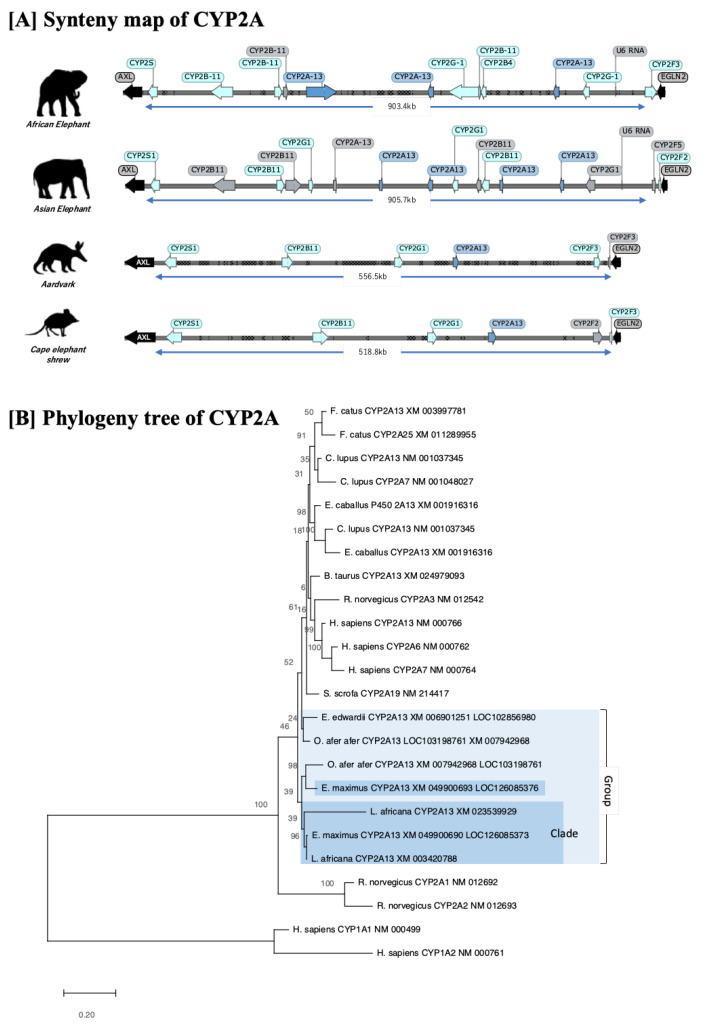
Synteny map and phylogeny tree of CYP2As gene of 12 mammals. (**A**) Synteny of CYP2As was conserved between AXL and EGLN2 genes. The dots in the synteny map indicate where N (Non-coding) is coded in the database. The blue arrows indicate the full coding region (intact gene) in CYP2A, while gray arrows indicate pseudogenes or partial genes. The light blue arrows are for other CYP2 families conserved in the same region as intact genes. (**B**) A phylogenetic tree of CYP2A in 12 selected mammals. CYP2As of both elephant species were in the same elephant-specific clade. CYP2As of close relatives such as the aardvark and Cape elephant shrew are located closer to the clade of other mammals than the elephant clade. Manatee’s CYP2A (XM_004390843/LOC101351737) was found in NCBI, but it was excluded from the analysis based on our analysis sequence, making up less than 50% of the entire length of the coding region.

**Figure 3 animals-13-01939-f003:**
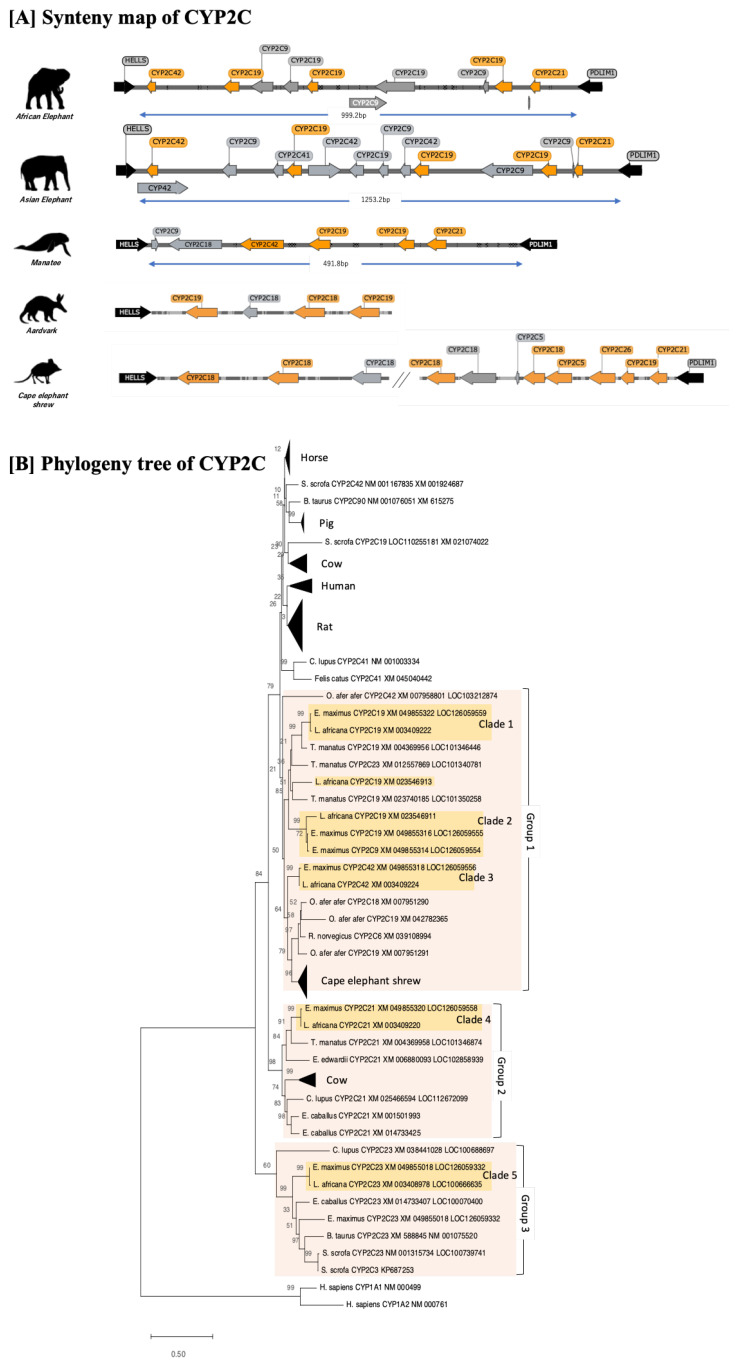
Synteny map and phylogeny tree of CYP2Cs gene of 12 mammals. (**A**) Synteny of CYP2Cs was conserved between HELLS and PDLIM1 genes. The orange arrows indicate intact genes in CYP2C, gray arrows indicate a gene annotated as pseudogenes in NCBI or partial genes based on our analysis. Aardvark sequence was retrieved differently from both sides of HELLS and PDLIM1 genes due to the low quality of sequence connection. (**B**) A phylogenetic tree showing that CYP2Cs of elephants can be separated into five different groups; CYP2C19 (clade 1), another group of CYP2C19 (clade 2), CYP2C42 (clade 3), CYP2C21 (clade 4), and CYP2C23 (clade 5). The clades can be further grouped into three broad categories. The first category includes clade 1, clade 2, and clade 3, which, except for CYP2C6 in rats, duplicated and developed almost independently in Afrotheria. The second category, containing clade 4, shows that genes were conserved in cattle, horses, dogs, Cape elephant shrews, and manatees. Finally, clade 5 diverged earlier as it was located far from the other CYP2Cs. This gene was also conserved and showed orthologs in cattle, horses, pigs, rats, and dogs.

**Figure 4 animals-13-01939-f004:**
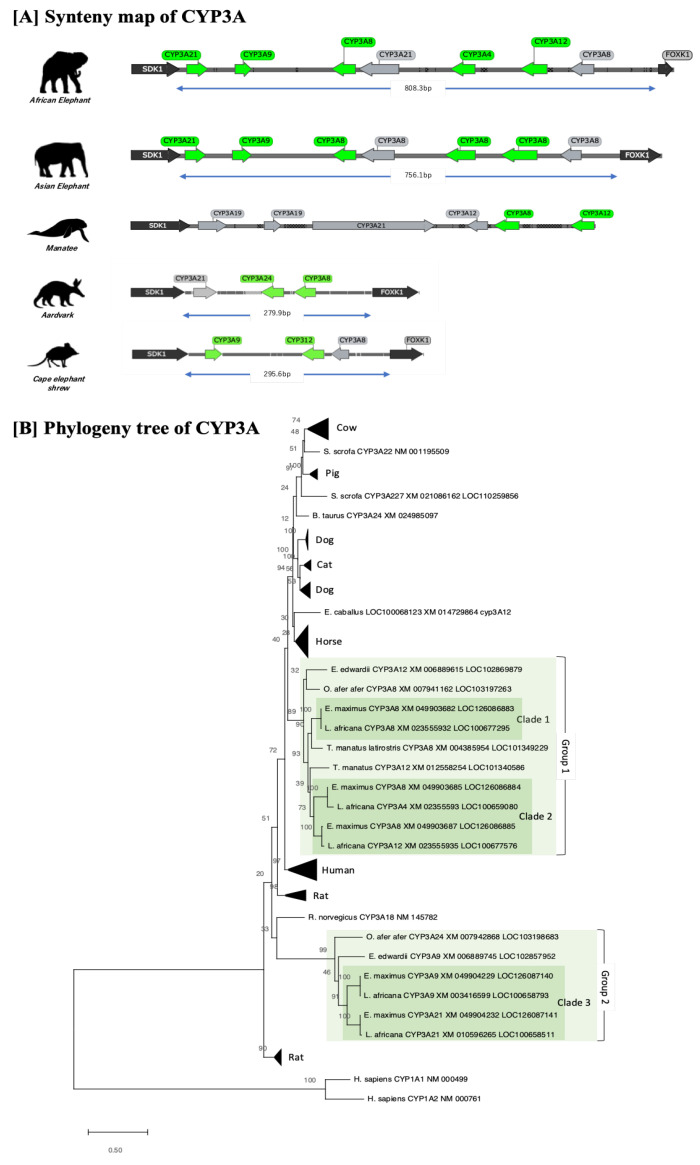
Synteny map and phylogeny tree of CYP3A genes of 12 mammals. (**A**) Synteny of CYP3As was conserved between SDK1 and FOXK1 genes. The green arrows indicate intact genes in CYP3As, while gray arrows indicate pseudogenes or partial genes. CYP3A-conserved loci were similarly conserved between African and Asian elephants. (**B**) Phylogenetic tree of CYP3A in 12 selected mammalian species. Elephant CYP3A was classified into two groups, with group 1 genes, including clade 1 and clade 2, clustered within clades of other mammals, and elephant-specific replication was observed within this clade. Group 2, including clade 3, formed a cluster of only Afrotheria species, which was isolated from other mammalian clades.

**Figure 5 animals-13-01939-f005:**
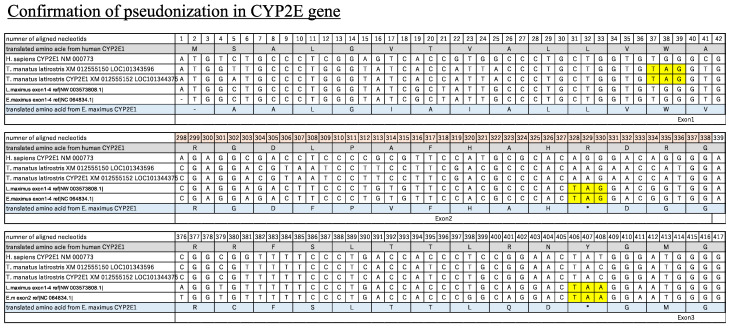
Loss of CYP2E genes in African and Asian elephants was confirmed by MEGAX alignment. CYP2E gene sequences were visualized and compared to confirm pseudogenes. Human CYP2E11 (NM_000773) were referenced for nucleotide number of aligned genes. Different exon regions were described and highlighted in the manuscript. In addition, human and Asian elephant amino acids are listed at the top and bottom of the table. As a result, in all sequences, stop codons (∗) were found. Stop codons were found in Exon2 (#343-345nt) and Exon3 (#421-423nt).

**Figure 6 animals-13-01939-f006:**
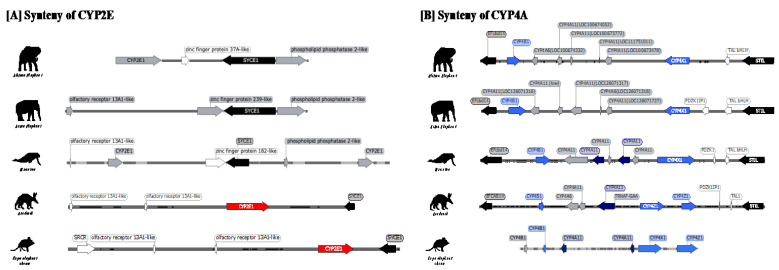
Synteny map of deficient gene CYP2E and CYP4A. (**A**) Synteny of CYP2Es was confirmed in between SYCE1 and SCART1 genes. In elephants and manatees, while SYCE1 gene was detected in these three species, SCART1 were not detected in any of them. Full coding region of CYP2E1 was only detected in Aardvarks and Cape elephant shrews, which is shown by the red arrows. Asian elephant CYP2E genes were not detected by the BlastN search by human CYP2E1 query, despite how we further conducted the query search in WGS region. (**B**) Synteny of CYP4As was confirmed in between EFcbd14 and STIL genes. Dark blue arrows indicate intact genes in CYP4A, while light blue arrows indicate CYP4 families conserved in same region. The loci were conserved in almost all the Afrotheria species; however, all the annotated CYP4As was either pseudogenized or partial genes in both African and Asian elephant. The loci were conserved in almost all Afrotheria species; however, all the annotated CYP4A genes were either pseudogenized or partial genes in both African and Asian elephants.

## Data Availability

Not applicable.

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
