# Peer review of "A Comparative Genomic and Phylogenetic Investigation of the Xenobiotic Metabolism Enzymes of Cytochrome P450 in Elephants Shows Loss in CYP2E and CYP4A"

_animals, 2023, doi:10.3390/ani13121939_

Round 1
Reviewer 1 Report
Authors characterized the CYP gene family in elephant by comparing 12 different mammalian species and uncovered unique features in elephant. Although they used only published data and there is no experimental validation, the research methods was straightforward and clear. In addition, the contents and results were well represented. So I suggested this manuscript for publication.
Author Response
We sincerely thank you for your time in peer review. We also appreciate your positive comments on our paper.
Reviewer 2 Report

Please correct the typos throughout the manuscript.
Author Response
Reviewer: 2
- (L68, L71, and many others) Copy number variation (CNV) refers to the varied copes of the same gene, not the varied numbers of the homologous genes. Please reword it throughout the manuscript.
We appreciate your review regarding our manuscript. According to your review we reword the “Copy number variation (CNV)” to “isoform numbers” throughout the manuscript.
- (L76-77) Please describe why the species were selected (aardvark, manatee, Cape elephant shrew, horse, rat, dog, cat, pig, cattle, and human).
Thank you for your comments on this part. The explanation of the species selection is added as following sentences:
L73-78
In this study, we aim to classify the existing xenobiotic metabolic types of CYP, specifically CYP1A, 2A-E, 3A, and 4A, in African elephants and Asian elephants. We achieved this by comparing annotated gene data for each gene assembly of three closely related species from Afrotheria (aardvark, manatee, and Cape elephant shrew) and various well-known mammals (horse, rat, dog, cat, pig, cattle, and human) whose xenobiotics have been studied and whose sequence annotations are known.
- (L131-132, and others) There is confusion between gene and mRNA. For example,“genes” should be mRNA because GenBank sequences with NM or XM are reference sequences of mRNAs, not genes. Please make the correction throughout the manuscript.
Thank you for your clarification. We have made the changes you suggested by replacing the word 'genes' with 'mRNA' in L135. Additionally, we have included the genomic name of the sequence in the following sentences;
L132-136
To this end, reference sequence of human CYP2E1 (Gene ID: 1571 / NM_000773) and CYP4A11 (Gene ID: 1579 / NM_000778) and manatee CYP2E1 (LOC101343596 / XM_012555150, LOC101344375 / XM_012555152) and CYP4A11 (LOC101345950 / XM_012554256, LOC101346203 / XM_004371774) mRNAs were used as a query sequence for the NCBI BLAST searches.
- (L178) total gene length -> the entire length of the coding region?
Thank you for this correction. As you mentioned in this sequence, we didn’t include the exon and intron, which “total gene length” might not be the appropriate word. The words are changed to “the entire length of the coding region” in both L182 and Fig2 legends.
- (L181-184) The entire length of the gene cluster does not necessarily reflect the numbers of gene duplication/loss. Please reword it.
We deeply appreciate this comment. We understand that the previous sentence may have caused some confusion regarding the factors influencing the number of gene duplications or losses in the entire cluster. Taking your comment into consideration, we have revised the sentences as follows:
L182-189
In the synteny maps of Afrotheria (Figure 2A), elephant show a greater length of the CYP2A coding region and significant gene expansion, including pseudogenes compared to the other two species in the clade (aardvark and Cape elephant shrew). In particular, the region length between the AXL and EGLN2 genes in both elephants was about 2-fold longer than in the aardvark and Cape elephant shrew. Although the entire length of the gene cluster may not directly reflect the numbers of gene duplication/loss, it does indicate that these loci are variable among Afrotheria species.
- (L203 and others) functional gene -> gene with the full coding region. One cannot say that gene is functional unless the gene is confirmed to be functional experimentally. Even some CYP genes contain the full coding region, but lack metabolic function. Please make the correction throughout the manuscript (especially many figure legends).
Thank you for this correction. The replacement of the term "functional gene" with "gene with the full coding region" provides a clearer description. Additionally, we use the word "intact gene" for alternative to provide clear explanation of this term. These modifications will help improve the clarity and accuracy of the manuscript.
Fig 2 and others
The blue arrows indicate for full coding region (intact gene) in CYP2A while gray arrow indicate for pseudogene or partial gene.
- This study provides information on the numbers of the genes in the same CYP subfamily, integrity of their coding regions, and their sequence identities to other genes, but one needs to investigate tissue expression and metabolic function of these CYPs to understand their importance as enzymes. Please describe the limitation of this study in Discussion.
We appreciate your review regarding our manuscript. About the needs to investigate tissue expression and metabolic function of these CYPs, we also consider this point as important issues. Therefore, we added the section in Discussion, mentioning the limitation of the data,
L460-472
The study provides initial insights into the presence and genetic characteristics of CYP subfamilies in African and Asian elephant, as well as other mammalian species. However, it is important to acknowledge the limitations that gene structure may not directly indicate functional enzymatic activity or metabolic function. Future research should focus on experimental investigations to determine tissue-specific expression patterns and metabolic activities of these CYPs. The study was also constrained by limitations in the quality of genome assemblies, which made it challenging to analyze certain CYP genes. Nonetheless, the findings highlight the presence of elephant specific metabolic mechanisms and caution against extrapolating physiological characteristics across species in drug use. Continued evaluation of this genomic data will play a crucial role in understanding species differences and further investigating the roles of these CYPs in processes such as xenobiotic metabolism, environmental adaptation, and other physiological functions in elephant.
- This reviewer recommends to include information on kockout mouse (CYP2E1, CYP4A) in Discussion.
We greatly appreciate your attention to the importance of discussing the deficient/knockout mouse models for both CYP2E1 and CYP4A. It is crucial to address the impact of these gene deficiencies on homeostasis and the development of diseases. Therefore, we have added a section in the Discussion to address these aspects as following:
L437-459
Furthermore, studies investigating the physiological dysfunction of the CYP2El gene in knockout mice showed no observed phenotypic abnormalities [66]. These findings indicate that CYP2E1 is not essential for normal development or physiological homeostasis, and the loss of CYP2E1 does not affect the expression of other CYPs. In contrast, the knockout mice with disrupted Cyp4a10 and Cyp4a14 genes exhibited a hypertensive phenotype [67–69]. These studies revealed that the targeted genes, Cyp4a10 and Cyp4a14, were not directly responsible for the hypertensive phenotype, but instead regulated the expression of other P450 genes involved in hypertension. Although the deficiency of CYP2E1 did not show an effect on physiological characteristics, it has been associated with nonalcoholic fatty liver disease (NAFLD). NAFLD is a disease associated with obesity, diabetes, and metabolic syndrome, and the progression of NAFLD can lead to the development of nonalcoholic steatohepatitis (NASH). CYP2E1 and CYP4A, which are abundant enzymes, are associated with this NASH [70]. CYP2E1 has been reported to play a crucial role in promoting the development of NASH by initiating lipid peroxidation through increased production of reactive species. However, studies using Cyp2E1 knockout mice showed that NASH development was not prevented and there was no reduction in microsomal NADPH-dependent lipid peroxidation. Meanwhile upregulation of CYP4A10 and CYP4A14 was observed in these mice, indicating their potential com-pensatory roles. To further understand the role of CYP4A in this pathway, Cyp4A knockout mice were used, revealing the activation of CYP4As as an alternative pathway for producing reactive oxygen species (ROS) [71]. These results suggest that the absence of both CYP2E1 and CYP4A in elephant may be associated with lower generation of ROS, and lower incidence of certain diseases such as NASH. [70,71]
Minor points:
- (L77) examine -> examined
- (L129) search -> searches
- (L96) BLAST search was done by default or specific parameters? Please specify.
This was done by the default parameter.
- (L102) coded CYP2As annotated genes -> Please reword it.
Reword to “Existing annotated CYP2As genes”
- (L103) sequences -> which sequences? Coding sequences?
Explain by adding a word “Coding sequences.”
- (L104) expressed -> contained.
- (L138) referencing sequence -> reference sequence
- (L139) CDS -> Please spell out if this is the first appearance.
- (L159) pseudonymized -> pseudogenized.
- (L229) were -> was.
- (L294) Orange Highlighted areas -> Please reword it.
Reword the sentence as “Different exon regions were described and highlighted in the manuscript.”
- (L313) alternation pseudogene -> What does this mean? Please reword it.
Reword the sentence as “were the pseudogene”
- (L319) A part from -> Apart from.
- There are other typos. Please correct the typos throughout the manuscript.
We again deeply appreciate your comprehensive and critical review thorough entire our manuscript.